# Perceptions, awareness on snakebite envenoming among the tribal community and health care providers of Dahanu block, Palghar District in Maharashtra, India

Itta Krishna Chaaithanya[1,2☯], Dipak Abnave[1☯¤a], Himmatrao Bawaskar[3], Ujwal Pachalkar[1], Sandip Tarukar[1], Neha Salvi[1], Prabhakar Bhoye[4], Arun Yadav[4¤b], Smita D. Mahale[1,2], Rahul K. Gajbhiye[1,2]*

1 Model Rural Health Research Unit, Department of Health Research, Government of India, Dahanu, Maharashtra, India, 2 ICMR-National Institute for Research in Reproductive Health, Mumbai, Maharashtra, India, 3 Bawaskar Hospital and Clinical Research Center, Mahad, Raigad, Maharashtra, India, 4 Sub District Hospital, Dahanu, Maharashtra, India

☯ These authors contributed equally to this work.
¤a Current address: Tata Institute of Social Sciences, Tuljapur, Maharashtra, India
¤b Current address: Directorate of Health Services, Arogya Bhavan, St. George Hospital Campus, Fort, Mumbai, Maharashtra, India
* gajbhiyer@nirrh.res.in

**Data Availability Statement:** All relevant data are within the paper and its S1 Fig, S1 and S2 Text, S1–S3 Tables files.

## Abstract

### Introduction

India has remarkably the highest number of snakebite cases contributing to nearly 50% of the global snakebite deaths. Despite this fact, there is limited knowledge and awareness regarding the management practices for snakebite in the Indian population. The study aimed to explore the knowledge, awareness, and perception of snakes and snakebites, first aid, and treatment amongst the community and the frontline health workers in a tribal block of Dahanu, Maharashtra, India.

### Methods

A cross-sectional study was carried out from June 2016 to October 2018 in the Dahanu Block, Maharashtra. Perceptions, knowledge, awareness, and first-aid practices on the snakebites among the community were studied through focus group discussions (FGDs). Semi-structured questionnaires were used to assess the knowledge, awareness, and experience of the traditional faith healers, snake rescuers, frontline health workers on the snakebites and their management. A facility check survey was conducted using pre-tested questionnaires for different levels of the government health care facilities.

### Results

Most of the tribal community was aware of the commonly found snakes and their hiding places. However, there was inadequate knowledge on the identification and classification of venomous snakes. Belief in a snake god, the perception that snakes will not come out during

**Funding:** This study received support from Indian Council of Medical Research (www.icmr.nic.in) in the form of a grant (Tribal/113/2016-ECD-II) awarded to RKG. No additional external funding was received for this study.

**Competing interests:** The authors have no competing interests to declare.

thunderstorms, change in taste sensation, the ability of tamarind seeds or magnet to reduce the venom effect were some of the superstitions reported by the tribal community. The application of a harmful method (Tourniquet) as the first aid for snakebite was practiced by the tribal community. They preferred herbal medicines and visiting the traditional faith healers before shifting the patient to the government health facility. The knowledge on the ability to identify venomous snakebites and anti-venom was significantly higher amongst nurses and accredited social health activists (ASHAs) than auxiliary nurse midwives (ANMs) and multi-purpose workers (MPWs) (p < 0.05). None of the traditional faith healers; but nearly 60% of snake rescuers were aware of anti-venom. Fifty percent of the medical officers in Dahanu block did not have correct knowledge about the Krait bite symptoms, and renal complications due to the Russell viper bite.

## Conclusions

Inappropriate perception, inadequate awareness, and knowledge about snakes and snakebites may predispose the tribal community to increased risks of venomous snakebites. Unproven and harmful methods for snakebite treatment practiced by the community and traditional faith healers could be dangerous leading to high mortality. Therefore, a multi-sectoral approach of community awareness, mapping of vulnerable populations, capacity building of health care facility, empowerment of health care workers (HCWs) could be useful for reducing the mortality and morbidity due to snakebite envenoming in India.

## Introduction

Snakebite is a major public health problem affecting an estimated 5.4 million people per year with up to 2.7 million envenomings [1, 2]. Many snakebite victims, mostly in developing countries, suffer from long-term complications such as deformities, contractures, amputations, visual impairment, renal complications, psychological distress [3]. India is considered to be the most affected country with an estimated 1.2 million snakebite deaths (average of 58,000 per year) from 2000 to 2019 [4]. Out of these, a very small proportion of snakebite cases were managed at government hospitals as the majority of the patients were referred to traditional faith healers [5, 6].

In a neighboring country, Nepal, 80% of the snakebite deaths occurred either in villages or during transport to the health care facility [7]. These observations highlight an urgent need to understand the healthcare-seeking behavior of the community for snakebites.

Snakebite is an occupational hazard affecting agricultural workers, farmers [8]. Working in the fields, fetching potable water, going to school or outdoor activities without footwear, defecation in an open field or outdoor toilet are some of the activities associated with a higher incidence of snakebites in rural and tribal areas [9, 10]. Inappropriate perception, the practice of unproven traditional methods, and inadequate knowledge about snakes and snakebites may increase mortality due to snakebite envenoming [11, 12]. However, most of these deaths are preventable, and hence community awareness is crucial. The community should be aware of the occupational risks and simple, cost-effective measures that can prevent a snakebite. There is a need to increase community awareness on the prevention of snakebites, bring behavioral changes for reduction of occupational risk, empower the community on first aid skills, and early transfer of snakebite patients to the nearest health facility. Delay in anti-venom

administration, inadequate administration of the initial dose of anti-venom, absence of tracheal intubation, and ventilation by a bag valve mask or artificial ventilator could lead to higher mortality [10, 13]. Our earlier study reported a case fatality rate of 4.5% due to snakebite envenoming at Sub District Hospital (SDH), Dahanu, Maharashtra [14]. The majority of the snakebites (66%) were reported in younger and earning members of the family leading to a substantial financial burden. There was no awareness and prior training on the management of snakebites, no uniform protocol for the treatment and there was irrational use of the intradermal anti-venom test in the tribal region of Maharashtra [14, 15]. Therefore, the capacity building of healthcare workers (HCWs) is essential for reducing the deaths due to snakebite envenoming.

India, one of the largest low-middle income countries (LMIC), has around 8.6% of the tribal population, 705 different tribes with 75 tribes classified as particularly vulnerable tribal groups (PVTG) inhabiting a significant part of the underdeveloped areas of the country [16]. The majority of the tribal population (93%) lives in rural and hilly areas and is mainly engaged in agriculture and allied activities. There exists a huge gap between tribal and non-tribal populations concerning healthcare [16]. Dahanu block (19.97°N 72.73°E) in the Palghar District of Maharashtra State is one of the tribal blocks with a higher tribal population (~70%) [17]. Dahanu block has a tropical climate with a coast of the Arabian Sea on the west and Sahyadri Hills on the east. Different tribes are inhabiting the Dahanu block namely Warli, Kokana, Mahadev Koli, Malhar Koli, Dhodi, Katkari engaged in different occupations and follow different cultural practices. Warli tribes remain quite unassimilated from the rest of India and maintain their dress style, customs, religion, and ceremonies. Warlis live in hilly areas and their major occupation is hunting; while some are engaged as agricultural laborers. Katkari tribes are classified as PVTG and are engaged in various livelihood activities including production and sale of catechu, charcoal, firewood, and other forest products, hunting of small mammals and birds. They eat rodents including the black rats and greater bandicoot rats. Kokana, Mahadev koli, Malhar Koli and Dhodi tribes are mainly engaged in agricultural work [18]. Risk factors such as environment, occupations, wrong perceptions, and inadequate knowledge on snakes and snakebites may predispose the tribal population to increased risk of snakebite deaths. Therefore, the present study was undertaken to understand the perception, awareness, and knowledge of snakebites, prevention, first aid practices, and healthcare-seeking behavior of the community for snakebite treatment. The aim was also to explore the awareness, knowledge, and management practices for snakebites among the traditional faith healers, snake rescuers, and HCWs.

## Methods

### Study design and setting

This was a cross-sectional study conducted from June 2016 to October 2018 in the Dahanu block of Palghar district in Maharashtra, India. The Dahanu block has 70,742 households with 49.6% males and 50.3% females. Male and females have a literacy rate of 69.9 and 50.6 respectively [17]. SDH Dahanu has a capacity of 100 beds and has an intensive care unit (ICU), pediatrics, gynecology out-patient, and indoor facilities and caters to 51,000 population. SDH Kasa covers about 50,000 population with a capacity of 50 beds. Rural hospital (RH) Vangaon covers a population of 15,000 with 30 beds (Fig 1). Eighteen focus group discussions (FGD) were conducted from July 2017 to December 2017 to facilitate in-depth discussion to understand perception, awareness, and knowledge about snakes, snakebites, prevention, first aid practices, and healthcare-seeking behavior of the community for snakebite. The study was reported using COREQ criteria (see S1 Text).

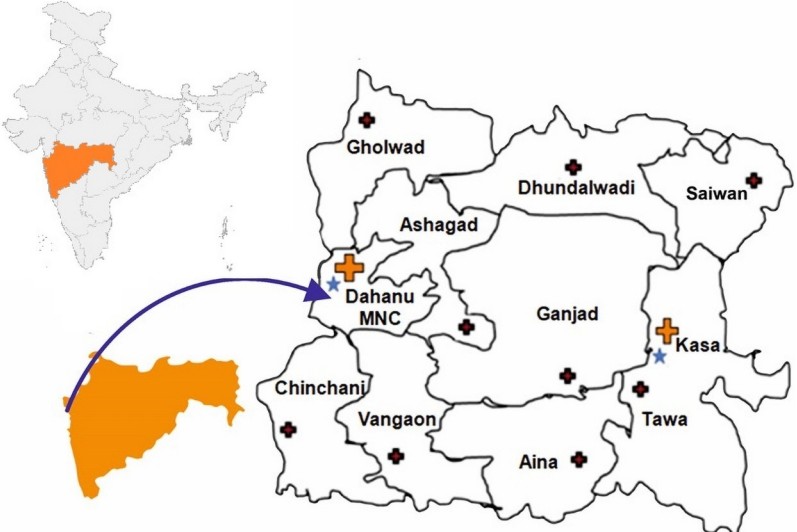

**Fig 1. The geographical location of Dahanu block, Palghar District, Maharashtra, India.** Map not to scale.

**Data collection tools.** The FGD guide was developed and adapted by our research team of public health experts, social scientist, and medical social workers (See S2 Text). All of them were proficient in the vernacular Marathi language. The draft version of semi-structured questionnaires for traditional faith healers, snake rescuers, frontline health workers, medical officers was evaluated independently by a panel of experts. The expert panel included experienced researchers including physicians, public health researchers, nursing staff, social scientist, medical officers working in the government healthcare facilities with adequate experience in snakebite management. Some of the questions were open-ended to capture the different perceptions, practices, and knowledge of the traditional faith healers, snake rescuers, frontline HCWs. The panel members reviewed the questionnaires and provided feedback for revision. The revised questionnaires were then validated by pilot testing at SDH Dahanu and primary health center (PHC) Ganjad in the study area. The research team also included one master's in social work (MSW), a native of the tribal area to facilitate and help the team to understand the socio-cultural issues. All 18 FGDs were carried out by the same facilitator with the help of two trained MSWs. Major themes explored in the FGD were:1) Awareness and knowledge about snakes and snakebites, 2) Perception and health-seeking behavior and 3) Awareness about first aid procedures for snakebites.

**Recruitment, sampling, and data collection.** FGDs were held separately for males and females for each PHC in the study area. The selection of the male and female participants for FGD was according to the convenience sampling method. Each FGD had approximately 8–16 participants and the participation was voluntary. Each FGD session was conducted for no longer than 60 minutes. The information about FGD participation was given to the community members through frontline HCWs and community leaders. All FGDs were facilitated by the social scientist who communicated with the frontline HCWs and community leaders for finalizing the date and time of FGD. Once the date of the FGD was finalized, the nearest government healthcare facility was selected as the venue in consultation with frontline HCWs and community leaders. The social scientist contacted the frontline HCWs and community leaders over the telephone a day before the scheduled FGD for reminding them and re-confirming the participation of the invited participants from the community. On the day of the FGD, the

research team carried the printed consent forms in vernacular Marathi language, audio recorder, stationery, and ensured the necessary logistic arrangements for conducting the FGD. The social scientist was assisted by two MSWs for writing the notes and recording the discussion. A copy of the participant information sheet and informed consent form was shared and explained in vernacular language (Marathi) to ensure voluntary participation. After obtaining the written informed consent, the social scientist assigned the responsibility to two MSWs for taking notes, monitoring the audio recording, and also checking the timer for keeping the track of time during the FGD. With the permission of study participants, the focus group discussions were audio-recorded and backed up by the field notes. The social scientist facilitated the discussion on perception, knowledge, awareness on snakes and snakebites, prevention methods, first aid practices, healthcare-seeking behavior of the community for snakebites. The social scientist encouraged the participants to explore the discussion topics in-depth and allowed them to raise their issues.

In the trial focus group discussions, it was observed that some of the community members were reluctant to share their age in front of other participants. It may be possible that some of the tribal community members may not know their exact age. Therefore, we could not record the age of the participants. However, the range of ages among participants regardless of gender was very broader from 18 to 60 years. There was a good response for participation in FGD. A total of 202 community members [Males 90 (44.5%) and Females 112 (55.4%)] participated in a total of 18 FGDs conducted for 9 PHCs representative of the tribal and non-tribal populations in the study area. Twelve participants were initially contacted but did not attend the FGD due to personal reasons. There were no dropouts during the FGD sessions. We tried to include participants reflecting a wide range of ages, community leaders, and members, older and younger to avoid any bias. Three trial FGDs conducted during the training were excluded from the present data analysis. During the discussion, if the participant used any tribal/other than local language, these points were written and further translated with the help of local snake rescuers (community volunteers who can handle live snakes).

Interviews of the traditional faith healers, snake rescuers, and frontline HCWs [Nursing staff, ASHA, ANM, and MPW] were conducted in the vernacular language (Marathi) using a semi-structured questionnaire. The research team visited health care facilities (Sub centers, PHCs, RH, and SDHs) in Dahanu block to conduct the interviews. The selection of HCWs (ASHA, MPW, ANM, Nurse) for conducting the interview was based on their willingness and availability. Interviews of a total of 96 HCWs were conducted representing the government health facilities in Dahanu block. Separate semi-structured questionnaires were used for different levels of HCWs. There was a separate questionnaire for traditional faith healers and snake rescuers. The semi-structured questionnaire included knowledge on venomous and non-venomous snakes, the symptoms of snakebite, preventive and first aid measures, traditional healing techniques, management practices, and knowledge on anti-venom.

**Facility survey.**   The Facility check survey was carried out in 2 SDHs, one RH, 9 PHCs, and 18 sub-centers under the Dahanu block with three different pre-tested questionnaires. The questionnaire included information on the availability of medical officers, Nursing staff, Pharmacists, basic investigations for snakebite, snakebite treatment facilities, availability of anti-venom, and other lifesaving drugs required for snakebite treatment, prior snakebite training details, etc. The selection of subcenters for the survey was by random sampling method. The questionnaires were administered to the head of the health care facilities.

**Capacity building of medical officers and frontline HCWs.**   A total of 40 medical officers from all the government health facilities in the Dahanu block attended the training on snakebite treatment which was conducted by one of the co-authors (HB) and included lectures, practical demonstrations, case studies, and in-depth discussions. A validated pre and post-test

questionnaires were administered to assess the knowledge, experience, and practices of snake-bite treatment. The questionnaire was based on the Standard Treatment Guidelines for snake-bite management, 2017 [19]. The test was conducted before the commencement and immediately on completion of the training to assess the knowledge gained. Periodic training was provided to the medical officers during the study period. The training was also provided to the frontline HCWs. A training manual in the vernacular language (Marathi) was provided to the frontline HCWs (S1 Fig). Copies of the quick reference guides and flyers (STG,2017) were provided to all medical officers.

**Data analysis.** Audiotapes of the full sessions of FGDs were transcribed. Transcripts were proofread and then translated into English by the experienced research staff to ensure accurate translation of the dialogue/statement of the participants. All transcripts were read independently by the investigators of this manuscript, who identified a list of themes and subthemes after reading a sample of interviews. For any doubts in translation, one MSW was able to go back and check the original transcripts in Marathi/local language and an experienced researcher was present and received simultaneous translation for all interviews. Themes and subthemes were identified based on the transcripts and coded the remaining transcripts by two experienced research staff. The thematic outline was subsequently tested with other samples of transcripts for modification.

Data from interviews of frontline HCWs, traditional healers, snake rescuers were entered into an excel sheet and analyzed further for frequency. Both pre and post-test data of medical officers training was entered in an excel sheet and the standard error of the difference (*P*-value) between the pre and post-test questionnaire was analyzed using MedCalc statistical software.

**Ethics approval.** The research study was conducted with the approval of the Institutional Ethics Committee of the Indian Council of Medical Research (ICMR)—National Institute for Research in Reproductive Health (NIRRH), Mumbai, India (D/ICEC/Sci-108/145/2016).

The study was also approved by the Director, Public Health Department, Government of Maharashtra, and Research Advisory Committee of Model Rural Health Research Unit (MRHRU), Dahanu, India.

## Results

### Focus group discussions (FGDs) among community members

**A. Awareness and knowledge about snakes and snakebites.** *a) Knowledge on common snakes found in the area.* A total of thirteen snake species were mentioned by FGD participants, which are commonly seen in the Dahanu block (Table 1). Majority of the participants were aware of the names of venomous and non-venomous snakes commonly observed in the Dahanu region (Fig 2).

*b) Hiding places of snakes.* The hiding places of snakes were reported in both residential and non-residential areas. The non-residential areas included agricultural farms mainly the rice plantations, rat and crab holes, under the green and dry grass, on trees, and under the tree in dry leaves. The storage of dry and green grass is a common activity in Dahanu block for animal fodder which was mentioned as a common hiding place for snakes. Similarly, the tribal community reported the Manilkara zapota (chikoo) plantation field as a common hiding place for snakes.

*c) Identification and classification of venomous snakes.* Participants mentioned the different methods such as the color, size, shape of the head, tail, and hissing sound for identification and classification of snakes. Indian Cobra was commonly identified as a venomous snake by the participants based on the hood mark. They further mentioned that number ten is inscribed on

**Table 1. The commonly found venomous and non-venomous snakes in the Dahanu block of Maharashtra, India.**

| S. No | Common Name | Scientific name | Local name used in the tribal community | Toxicity |
|---|---|---|---|---|
| 1 | Common krait | *Bungarus caeruleus* | *Manyar, Chuad* | Venomous |
| 2 | Russell's viper | *Daboia russelii* | *Ghonas, Kamblya* | Venomous |
| 3 | Indian Cobra | *Naja naja* | *Nag, Sap,* (*Dahaakadi*) Ten numbered snake, it was classified as black cobra | Venomous |
| 4 | Saw scaled viper | *Echis carinatus* | *Furdsa, Fursa,* | Venomous |
| 5 | Vine snake | *Ahaetulla nasuta* | *Harntol, Toli* | Mildly-Venomous |
| 6 | Indian rat snake | *Ptyas mucosa* | *Dhamin, Adela* or *Adhelwad* Sap | Non-Venomous |
| 7 | Red Sand boa | *Eryx johnii* | *Mahandulya, Madul* | Non-Venomous |
| 8 | Rock python | *Python sebae* | *Azgar* | Non-Venomous |
| 9 | Checkered keelback | *Xenochrophis piscator* | *Divad, Pansor, Panchita, Pansul* (Sweet water snake) | Non-Venomous |
| 10 | Bronze black tree snake | *Pseudonaja textilis* | *Rukhai,* | Non-Venomous |
| 11 | Striped keelback | *Amphiesma stolatum* | *Nanheti* | Non-Venomous |
| 12 | Trinket snake | *Coelognathus Helena* | *Taskar* | Non-Venomous |
| 13 | Worm Snake | *Indotyphlops braminus* | *Vala* | Non-Venomous |

the head of the cobra (Daha akadi in vernacular Marathi language). Some of the participants referred to it in vernacular Marathi language as "lehar" meaning a sign of 'U' or V on the snakehead or under the neck. However, there was inadequate knowledge on the identification and classification of other venomous snakes such as common krait, Russell's viper, and Saw Scaled Viper.

The participants mentioned that they could differentiate venomous and non-venomous snakes based on fang marks. They considered venomous snakebite based on two teeth bite marks. "*If more than two teeth bite marks were seen then they considered the bite as non-venomous*" (FGD, ID 07). *If there was more bleeding at the site of the snakebite, it was considered as venomous snakebite compared to no bleeding which was considered as non-venomous snakebite* (FGD, ID 12)

Two separate male respondents said "*People here in this community are not able to identify snakes by names or by watching the snake, and after the bite, they were able to identify either venomous or non-venomous* (FGD, ID 07,15) *and all snakes are venomous* (FGD, ID 13)

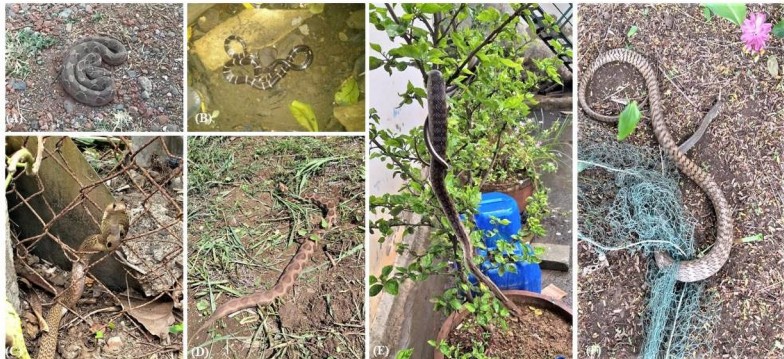

**Fig 2. Commonly found venomous and non-venomous snakes in Dahanu block, Maharashtra, India.** Venomous snakes: (A) Saw scaled viper (B) Common krait (C) Cobra (D) Russell viper; Non-venomous snakes (E)Trinket snake (F) Rat snake. Photo credit: Mr. Sagar Vijay Patel, Forest Department/Biodiversity Department, Dahanu, Maharashtra, India.

*d) Season and timing of snakebites*. Some of the FGD participants mentioned that more snakes are observed from May to October and most snakebites occur during the agricultural work in the monsoon season (FGD ID,03,15,18).

They further mentioned that during the monsoon snakes come out of their holes and encounter humans. (FGD, ID 01, 06).

**2. Perception and health-seeking behavior for snakebite.**    *a) Perception about snakes, snakebites, and treatment.* The tribal community believed that snakes residing in the agricultural field are sent by God to protect their farm and agrarian territory. They worship the *Snake God* (Nagdevta) by offering coconut fruit and milk to the Snake idol in the Temple. So, the tribal communities do not kill such snakes found in the agricultural field. On the contrary, snakes found in the residential areas are either caught by the snake rescuers or killed by the local community people. These are the two different perceptions reported in the Dahanu block.

A male respondent said, "*See, in our community, People worship cobra snake as god if they see in farms*

(FGD, ID01; FGD, ID07; FGD, ID17).

Some of the participants mentioned that snake takes revenge, especially Cobra. If people try to kill Cobra and if cobra survives then it will take revenge after some days. The tribal community believed that male and female snake species always live together (as life partners). If one of the partners is killed by a human; then the surviving partner will take revenge.

"*If you kill a male snake then the female snake will take revenge on you*"

(FGD, ID 09).

Community members believed that snake venom alters the taste sensation. They mentioned that green chilies or dry chili powder, salt, sugar are given to the snakebite victim to eat and if there is a loss of taste sensation, they considered the bite as venomous and *vice versa*.

A female respondent said, "*We give green chilies or dry chili powder, salt, sugar to the snakebite victims to eat, if they can identify the taste then it is a non-venomous bite but if they are unable to identify taste then it is venomous snakebite*"

(FGD, ID 10).

A female respondent said, "*After the bite if the victim cannot walk even 5 steps and death occurs immediately it was considered as venomous snakes*" (FGD, ID 08). Participants said, "*If we disturb the snakes. then the only snakes will bite humans*"

(FGD ID 01, 11).

Few of the community participants believed that "*If the snake bites once, a victim will not die, but if the snake bites the second time, the victim will die*"

(FGD ID 04).

The community members mentioned that "*Thunderstorm will affect the snake activities and the snake will not come outside from the hole during a thunderstorm*" (FGD ID 07). Some participants mentioned, "*If pregnant woman encounters a snake and if she looks at the snake then the snake will become blind*"

(FGD ID 08).

Such wrong beliefs and superstitions will predispose the community to a higher risk of snakebite deaths.

*b) Utilization of government health care services*. Participants mentioned that they preferred to visit nearby government healthcare facilities for the snakebite treatment. "First, we prefer for home remedies, if it's not cured then we visit government hospital (FGD, ID 03).

"*Few of them prefer local treatment with the herbal medicine (tamarind seed) to reduce the venom effect. Also, they approach a traditional healer if it is not recovered. If there is no improvement then they move to a nearby government hospital for treatment*"

(FGD ID 15, ID 17).

**3. Knowledge on prevention, first aid, and treatment of snakebite.**    *a) Prevention of snakebite*. The participants mentioned the use of household methods for the prevention of snakes. They prefer a sprinkling of locally available pesticide Dichlorodiphenyltrichloroethane (DDT) and insecticide Thimet powder around their houses and yards. They believed that DDT and Thimet powder have a strong odor and therefore the snakes cannot enter their house and will protect them from snakebites.

Female respondents said," *We use different methods such as DDT (pesticide), Thimet powder (insecticide) and Fish water (leftover water after cleaning fishes) sprinkling, and burning the scrap rubber tires, cow dung or firewood to prevent snakes*"

(FGD ID 02, ID04, ID 04, ID 08, ID 10).

Some of the participants mentioned that they are not aware of the prevention methods. "*We don't follow snakebite prevention methods as we are not aware of them*" (FGD ID 13).

Tribal communities in Dahanu commonly use dry cow dung, firewood as fuel for cooking purposes and store the dry/green grass for cattle feeding. However, these are the common hiding places for snakes. Some of the participants were aware of snakebite prevention measures such as keeping the cattle feed, firewood, dry cow dung above the ground level (FGD ID 17).

Some of the participants mentioned "*maintaining clean surroundings, use of torchlight and wooden sticks at night will prevent us from snakebites*"

(FGD ID 04, 05, 06, 13, 14).

The use of mosquito nets for the prevention of snakebites was mentioned by only one participant (FGD ID 05) suggesting that there is very limited awareness of snakebite prevention methods in the community.

*b) Knowledge on first aid for snakebites*. Use of a tourniquet or bandage above the site of the snakebite was reported as the first aid for snakebites by the participants (FGD ID 02,10,15). Overall, knowledge on the accurate use of first aid for snakebites was lacking in the majority of the study participants.

*c) Snakebite treatment at government health facilities*. Some of the participants were aware that, snakebite treatment is available in government hospitals. They further mentioned that the private hospitals do not admit snakebite cases as they do not have snakebite treatment facilities. Although some of the participants mentioned that they preferred government hospitals

for the snakebite treatment, however, they said: "*Government hospitals do not take care properly and do not respond in an emergency*" (FGD ID 05).

Some of the participants mentioned, "*Government hospitals do not provide treatment properly and also there is a long waiting time and non-availability of doctors during emergencies such as snake bites*". (FGD ID 03)

### Interviews of HCWs, traditional faith healers, snake rescuers

Among 96 HCWs, 38 were Nursing staff, 35 were ASHA workers, 11 were ANMs, and 12 were MPWs. Nearly 58% of HCWs were having job experience of $> 10$ years with 48% working in the Dahanu area for $>10$ years (S1 Table). The ability to identify venomous snakebites and anti-venom were significantly higher amongst nurses and ASHAs than ANMs and MPWs ($p < 0.05$). The details of the analysis of knowledge, awareness, and first aid for snakebites amongst various HCWs are shown in Fig 3. Out of the total 9 traditional faith healers interviewed, 67% (n = 6) were educated below the primary level and 33% (n = 3) were above the primary level. The majority of them were working as the soul faith healer (tantric). About 33% of the faith healers were treating snakebite cases for more than 10 years. Sixty-six percent of the faith healers mentioned that they were able to identify venomous and non-venomous snakes. None of the traditional faith healers were aware of anti-venom (Fig 4). The majority of them (78%) were sending the snakebite cases to a nearby government health care facility if they were unable to manage. The traditional faith healers were reluctant to share the information on what traditional methods/medicines they use for treating snakebite patients due to the perception of losing the effectiveness of their method if shared with others.

Sixty-six percent of the snake rescuers had experience rescuing snakes for more than 10 years. More than 40% of them were working in the Dahanu area for more than 10 years (S2 Table). All of them were aware and able to identify the venomous or non-venomous snakes

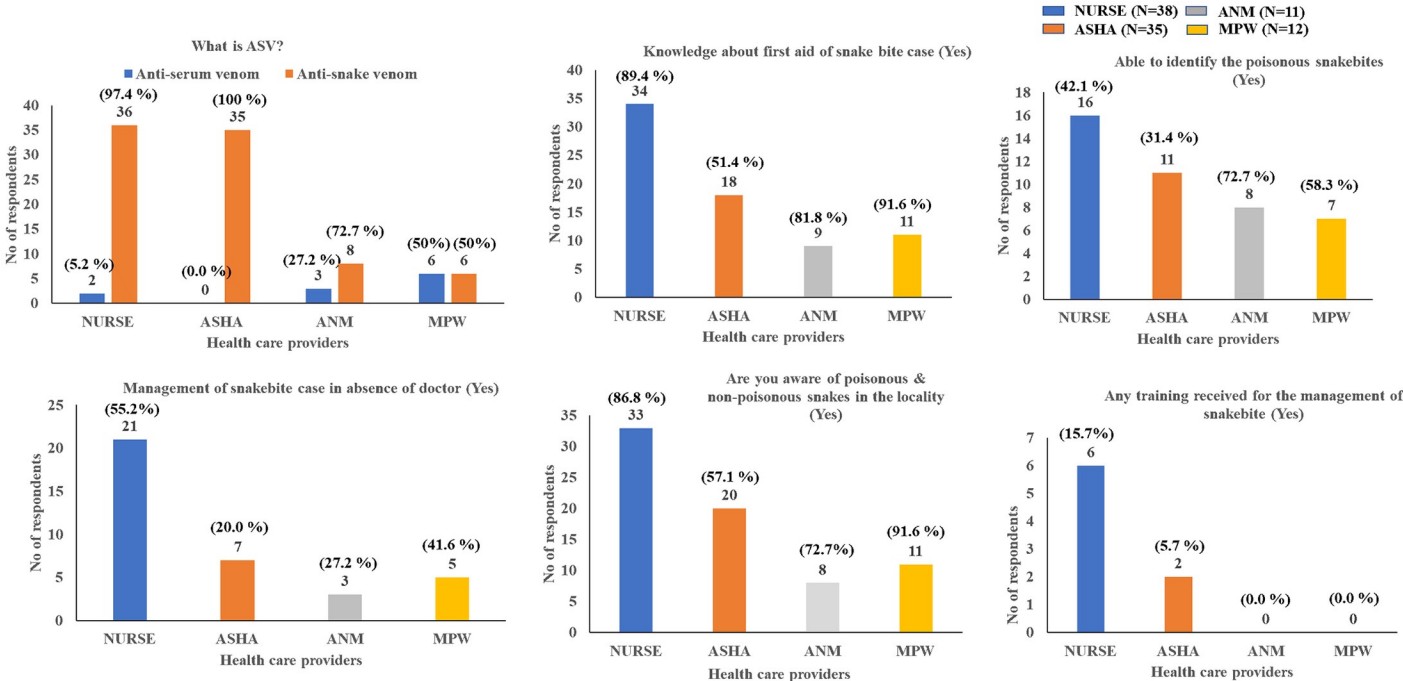

**Fig 3. Awareness, knowledge on snakebite and first aid practices amongst the frontline health care workers in Dahanu, Maharashtra.**

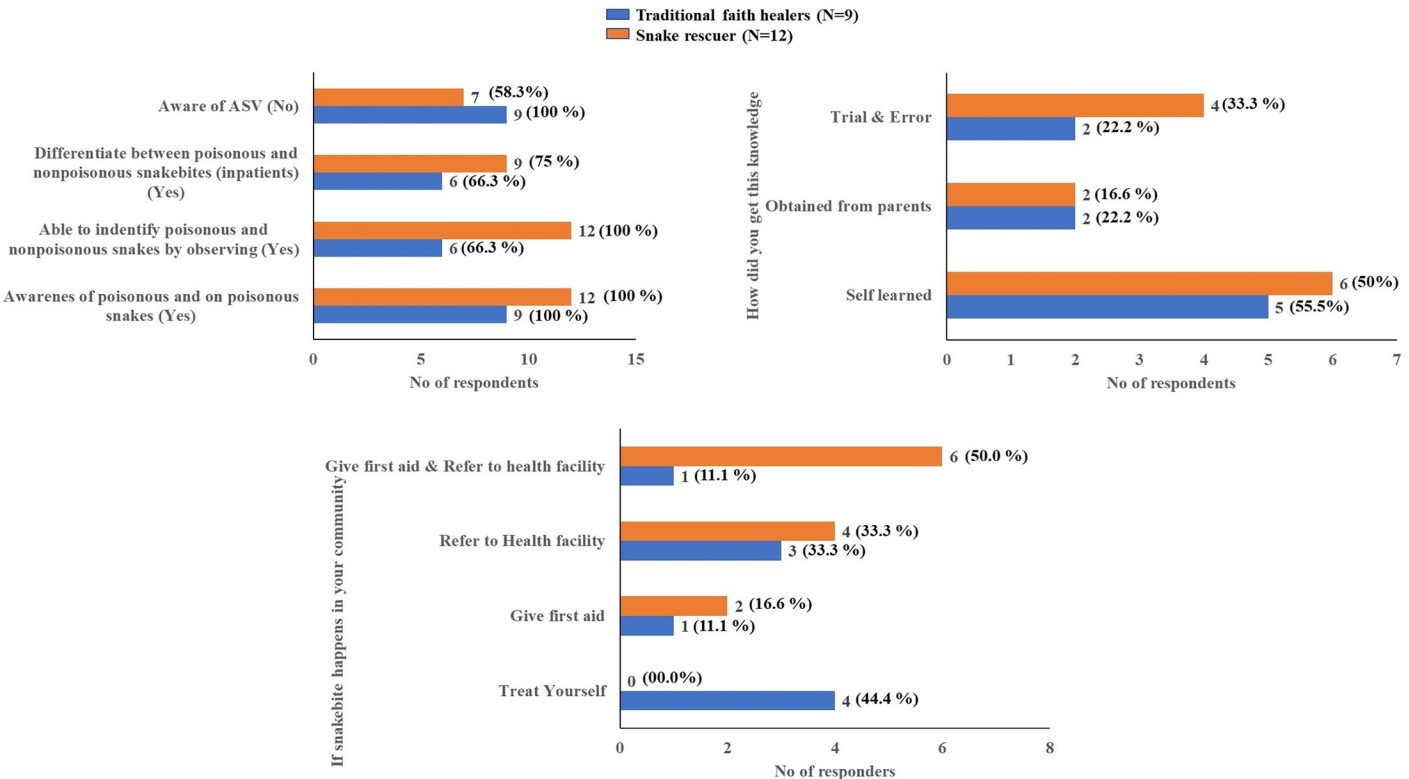

**Fig 4. Awareness, knowledge of snakebite and treatment practices amongst the traditional faith healers and snake rescuers.**

found in the study area. Fifty percent of the snake rescuers were aware of the first-aid practices and referred snakebite patients to the nearby government health care facility. About 42% of the snake rescuers had information about the availability of anti-venom in government hospitals for snakebite treatment (Fig 4).

## Facility survey in government health care facilities

Basic laboratory investigations (bleeding time, clotting time, complete blood count, etc.) were available in SDHs and RH. Anti-venom was found available in all SDHs, RH, and PHCs in the Dahanu block suggesting the regular supply of anti-venom by the state government. The bleeding time and clotting time testing facility were available only in 4 PHCs. All 18 SCs that were randomly selected were found operational in the government-owned buildings. For 56% of the SCs, the nearest PHC was less than 10 km whereas for, 44% of SCs, the distance to the nearest PHC was more than 10 km (20 km maximum). There was no IEC (Information, Education, and Communication) material available on the identification of venomous and non-venomous snakes, prevention, first aid, and treatment of snakebites in any of the government healthcare facilities in the study area.

## Training evaluation of medical officers

The majority (85%) of the medical officers were aware of the signs and symptoms of snakebites. Fifty-five percent of the medical officers were aware of krait bite symptoms (without local Pain or tissue damage) and only 45% knew about renal complications caused by Russells Viper (S3 Table). There was no prior awareness about the National snakebite management

protocol, (2009) Govt. of India or Standard Treatment Guidelines for snakebite management (STG,2017).

## Discussion

The study demonstrates inadequate knowledge, wrong perceptions, use of unproven methods for prevention and management of snakebites amongst the tribal community in Dahanu, India. The community had wrong perceptions on the identification of venomous snakes and snakebites. Belief in a snake god, ability of tamarind seeds or magnets to reduce the venom effect were some of the superstitions reported in our study. The community had a wrong belief that if anyone kills a male snake, then a female snake would take revenge on that person which was also reported in a study conducted in Nepal [20]. Such wrong perceptions and beliefs will make the tribal community at a higher risk of snakebite deaths.

Traditional knowledge of the local tribal community about the taste alternation in venomous snakebite needs further exploration as it has a scientific basis. There is evidence reporting alterations in smell and taste in envenoming by several snake species [21–23]. Further investigations are needed to understand whether these are central effects or due to peripheral cranial nerve involvement [24]. Although the majority of the tribal community is following unproven household methods (sprinkling of DDT, Thimet powder) for the prevention of snakebites, there was some awareness on the use of a torch, wooden sticks, and mosquito net. Wearing long rubber shoes with enclosed toes, use a mosquito net on a bamboo cot or bed above the ground level could offer protection from krait and cobra bites [10, 25, 26]. However, there are challenges in the implementation of preventive measures, both amongst the illiterate and highly educated class indicating the need for education and awareness right from the school level to the university [27]. WHO strategy for prevention and control of snakebite envenoming focuses on prevention of snakebite, provision of safe and effective treatment, strengthening health systems, and increased partnerships, coordination, and resources [28, 29]. Evidence generated from this study and other studies from different geographical regions would be useful for the effective implementation of the prevention and control program of snakebite envenoming.

The use of alternative and unproven methods for the treatment of snakebite patients is still followed in many countries including India [30, 31]. Although there are reports on the use of snake stone (black stone) to treat snakebites since ancient times, there is negligible scientific literature on its therapeutic efficacy. Chippaux *et al.* conducted a study to determine the therapeutic efficacy of black stones using an animal model but failed to show any therapeutic efficacy [32]. Since traditional remedies do not have any proven benefit in treating snakebite, it is recommended to avoid traditional first aid methods including black stones and alternative medical/herbal therapy [33]. In our study, the community preferred local treatment with tamarind seed or magnet to reduce the venom effect and approached the traditional faith healers. If the victim does not recover, only then they preferred to transfer the patient to the nearby government healthcare facility. Similar use of tamarind seed was also observed in *Bhil* tribes of Rajasthan, India [34]. The use of such unproven methods was associated with increased risk of bite wound infection, long duration of hospitalization for the management of snakebite victims [30, 35]. Therefore, the community should be educated to discard such practices and unproven methods for snakebite treatment.

There could be several reasons for seeking treatment from the traditional faith healers. These include: i) strong belief of rural and tribal communities in the traditional healers; ii) low cost for traditional healing; iii) non-availability of doctors; iv) shortage of anti-venom in public healthcare facilities, and iv) limited access to public transportation. Sometimes, medical

officers are not available at the public health facility especially during the night hours [36]. So, the snakebite victims are left with the option of going to the locally available traditional healer. Hence, we suggest developing a good understanding among the traditional faith healers and the public health departments. Traditional faith healers should be provided with proper training on detection of signs of envenoming for timely referral of patients to the nearest hospital for anti-venom treatment. Providing incentives to the faith healers for refereeing and accompanying the snakebite envenoming cases to the hospital can be a good strategy to reduce mortality. Towards this, efforts are ongoing in West Bengal, India to engage the traditional faith healers for timely referral to the nearest hospital for anti-venom treatment [37].

Despite the community's awareness of the availability of snakebite treatment at government hospitals, they were reluctant to visit the government hospitals. This could be due to the unavailability of doctors during night time, lack of awareness in the doctors regarding the proper dosage of anti-venom, and poor confidence level of the treating doctors in the primary health care level [38]. Elsewhere studies in India and other countries have also identified similar gaps such as the inability to identify systemic envenoming and administration of anti-venom [37, 38]. Linkages should be build up with community leaders and health care providers to gain the confidence of the community for timely referral to the nearest hospital having anti-venom treatment.

The availability of experienced HCWs in tribal and rural areas is an important aspect of the prevention and control of snakebite cases. In the present study, around 50% of the medical officers lacked the correct knowledge about krait bite symptoms and complications due to Russell viper bite. Around 60% of the medical officers did not know about essential laboratory tests done for the diagnosis of venomous snakebites. Eighty-five percent of the medical officers answered correctly that intravenous injection was the appropriate route for anti-venom administration. These findings were similar to those from the earlier studies [29, 39]. Before our study, no formal training on the management of snakebite was being provided to the medical officers in the Dahanu block. Therefore, despite the availability of anti-venom, most of the medical officers did not have confidence in administering the anti-venom mainly due to a fear of anaphylaxis reaction and lack of formal training.

Globally, around 50% of the poor population is completely or partially dependent on livestock for their livelihoods [40]. Domestic animals including cattle, goats, horses, sheep are an important part of the livelihoods of tribal communities in the Dahanu. These domestic animals are often affected by venomous snakebites, causing fatality rates of more than 47% in livestock [41]. Therefore, active engagement with communities to create awareness on the prevention of human snakebites will increase their awareness about snakebites among domestic animals [42] to protect them from snakebites. Therefore, knowledge and awareness of the venomous and non-venomous snakes, perceptions about snakes and snakebites are of paramount importance in reducing the burden of snakebites in India and other tropical countries. The WHO strategy on prevention and control of snakebites recommends a model of "One Health" which includes collaborations between human and veterinary healthcare systems [42].

## Study limitations

Our study had the following limitations. The study was carried out in the Dahanu block of Palghar District which may not be representative of the total tribal population in India. Thus, perceptions and awareness about snakebites could be region-specific and may not be representative of the entire tribal population in India. All FGDs and interviews were conducted in the vernacular language (Marathi) and later translated into English. Local snake rescuers were involved in the translations and interpretations of some of the words used by tribal

communities during FGDs. Hence, some interpretations might have been lost in translation. We could not record the age of the FGD participants and the study missed out on an age-wise data analysis. The pre-and post-test to assess the knowledge and management practices were carried out just before the commencement of training and immediately after the training of medical officers. Therefore, the results of the pre- and post-training survey might reflect what was taught/discussed during the training.

## Conclusions

The present study generated evidence to empower the community by increasing awareness on the prevention of snakebite, first aid, and appropriate treatment-seeking behavior. Culturally appropriate IECs should be developed for increasing the awareness and sensitizing the community for early referral of snakebite victims to the nearest health facility having anti-venom treatment. Further, large-scale studies on exploring the traditional knowledge and practices of the tribal communities should be undertaken on priority.

## Recommendations

1. To include snakebite management in the curriculum of training institutions of the state public health departments in India.

2. Mandatory short-term training of medical graduates during their internship and also as a part of the induction training on joining the state health services in India.

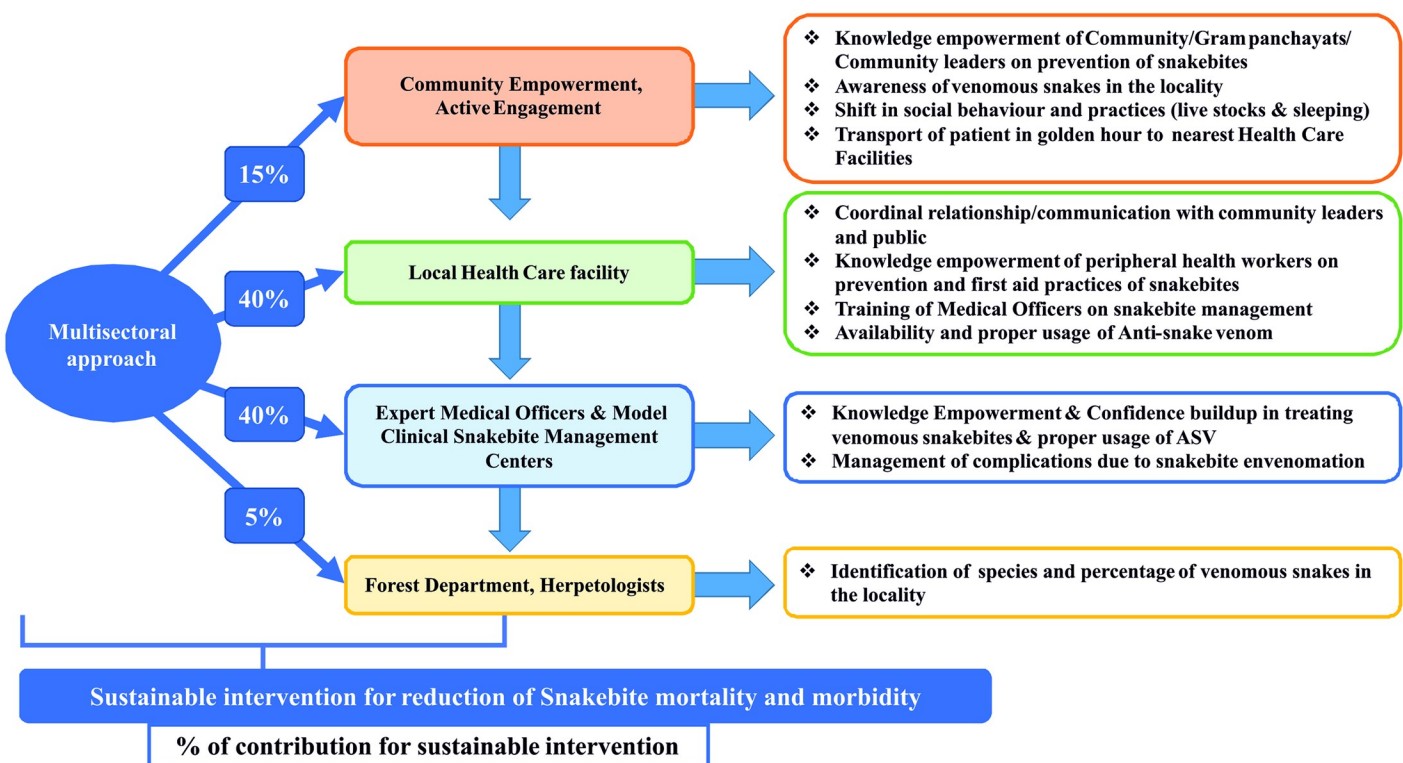

**Fig 5. Multi-sectoral model for reduction of snakebite mortality and morbidity in rural India.**

3. To develop a policy to ensure periodic in-service training on snakebite management as per the National snakebite management protocol.

4. Periodic review and evidence-based update of the National snakebite management protocol

5. To involve the program managers at national and state levels for successful implementation of the National snakebite management protocol.

## Implications of the present study

Global mapping of hotspots identified the most vulnerable populations to 278 medically important snake species responsible for the most severe outcomes of snakebite envenoming [43]. Based on these observations and our experience from the present study, we propose a multi-sectoral model which includes the community education, mapping of vulnerable populations to the most severe outcomes of snakebite envenoming, capacity building of local health care facility, empowerment of medical officers and HCWs on the management of snakebite as per National snakebite management protocol, transdisciplinary expertise including animal health, herpetology, forestry, anthropology, and education (Fig 5). The model could be useful for reducing the mortality and morbidity associated with snakebites in rural India as well as in other tropical countries.

## Supporting information

**S1 Fig. Training manual on snakebite envenoming for primary health care workers in the tribal region of Dahanu, Maharashtra, India.**
(TIF)

**S1 Table. Snakebite experience of frontline health care workers in Dahanu block, Maharashtra, India.**
(DOCX)

**S2 Table. Demographic details and awareness of snakebite amongst the traditional faith healers, and snake rescuers.**
(DOCX)

**S3 Table. Pre and post-training evaluation of medical officers.**
(DOCX)

**S1 Text. COREQ checklist.**
(DOCX)

**S2 Text. Focus group discussion guide.**
(DOCX)

## Acknowledgments

The authors are sincerely thankful to Dr. V M Katoch and Dr. Kiran Katoch for their guidance and motivation for the conceptualization and implementation of this study. The authors are thankful to Dr. Soumya Swaminathan, Dr. Raman Gangakhedkar, Dr. Harpreet, and the ECD Division of ICMR for facilitating financial assistance for the project. Dr. Ashoo Grover, Dr. Sangeeta Sharma is sincerely acknowledged for providing the final version of the Standard Treatment Guidelines of the Government of India. Dr. Satish Pawar, Dr. Mohan Jadhav, Dr.

Archana Patil, Dr. Sanjeev Kamble, Dr. Ratna Ravkhande, Dr. Shyam Nimagade, Dr. Umesh Shirodkar, Dr. Geeta Kharat, Dr. Santosh Gaikwad, Dr. Kanchan Vanere, Dr. Sanjay Bodade, Dr. Balaji Hengne, Dr. Mitesh Torankar, Dr. Abhijit Chavan and officials from Public Health Department, Government of Maharashtra are sincerely acknowledged for their support in the implementation of the study. The staff of SDH Dahanu and MRHRU, Dahanu are sincerely acknowledged for extending the support for the implementation of the study through MRHRU Dahanu. Dr. Yogeshwar Kalkonde and Dr. Taruna Madan are sincerely acknowledged for the critical review and assistance in editing the manuscript. Dr. A. R. Pasi and Dr. Ranjan Kumar Prusty are acknowledged for assistance in statistical analysis.

## Author Contributions

**Conceptualization:** Himmatrao Bawaskar, Rahul K. Gajbhiye.

**Data curation:** Itta Krishna Chaaithanya, Dipak Abnave, Ujwal Pachalkar, Sandip Tarukar, Neha Salvi, Rahul K. Gajbhiye.

**Formal analysis:** Itta Krishna Chaaithanya, Dipak Abnave, Rahul K. Gajbhiye.

**Funding acquisition:** Rahul K. Gajbhiye.

**Investigation:** Itta Krishna Chaaithanya, Dipak Abnave, Prabhakar Bhoye, Rahul K. Gajbhiye.

**Methodology:** Himmatrao Bawaskar.

**Project administration:** Itta Krishna Chaaithanya, Arun Yadav, Smita D. Mahale, Rahul K. Gajbhiye.

**Resources:** Prabhakar Bhoye, Arun Yadav, Smita D. Mahale.

**Supervision:** Neha Salvi, Smita D. Mahale.

**Writing – original draft:** Itta Krishna Chaaithanya, Dipak Abnave, Himmatrao Bawaskar, Ujwal Pachalkar, Sandip Tarukar, Rahul K. Gajbhiye.

**Writing – review & editing:** Itta Krishna Chaaithanya, Himmatrao Bawaskar, Smita D. Mahale, Rahul K. Gajbhiye.

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
