## [Decision Letter · Decision Letter 0]

15 Jan 2021

PONE-D-20-33739

Perceptions, awareness on snakebite envenomation among the tribal community and health care providers of Dahanu block, Palghar district in Maharashtra, India

PLOS ONE

Dear Dr. Gajbhiye,

Thank you for submitting your manuscript to PLOS ONE. After careful consideration, we feel that it has merit but does not fully meet PLOS ONE’s publication criteria as it currently stands. Therefore, we invite you to submit a revised version of the manuscript that addresses the points raised during the review process.

We look forward to receiving your revised manuscript.

Kind regards,

Ritesh G. Menezes, M.B.B.S., M.D., Diplomate N.B.

Academic Editor

PLOS ONE

Journal Requirements:

2. Please include your tables as part of your main manuscript and remove the individual files. Please note that supplementary tables should remain uploaded as separate "supporting information" files.

4. In the Methods, please discuss whether and how the questionnaire was validated and/or pre-tested. If these did not occur, please provide the rationale for not doing so.

5. Thank you for stating the following in the Funding Section of your manuscript:

"RG is an awardee of the DBT

 Wellcome India alliance clinical and public health intermediate fellowship (Grant no.

IA/CPHI/18/1/503933)."

"Initials of the authors who received Funding : RG, SM

Grant numbers awarded :

Tribal Health Research Forum, Indian Council of Medical Research and Department of Health Research, Ministry of Health and Family Welfare, Government of India for financial assistance (NIRRH/MS/RA/886/03/2020)

The full name of each funder

URL of each funder website

www.icmr.nic.in

Did the sponsors or funders play any role in the study design, data collection and analysis, decision to publish, or preparation of the manuscript?

NO"

6. We note that Figure 1 in your submission contain map images which may be copyrighted. All PLOS content is published under the Creative Commons Attribution License (CC BY 4.0), which means that the manuscript, images, and Supporting Information files will be freely available online, and any third party is permitted to access, download, copy, distribute, and use these materials in any way, even commercially, with proper attribution. For these reasons, we cannot publish previously copyrighted maps or satellite images created using proprietary data, such as Google software (Google Maps, Street View, and Earth). For more information, see our copyright guidelines: http://journals.plos.org/plosone/s/licenses-and-copyright.

6.1.    You may seek permission from the original copyright holder of Figure 1 to publish the content specifically under the CC BY 4.0 license. 

6.2.    If you are unable to obtain permission from the original copyright holder to publish these figures under the CC BY 4.0 license or if the copyright holder’s requirements are incompatible with the CC BY 4.0 license, please either i) remove the figure or ii) supply a replacement figure that complies with the CC BY 4.0 license. Please check copyright information on all replacement figures and update the figure caption with source information. If applicable, please specify in the figure caption text when a figure is similar but not identical to the original image and is therefore for illustrative purposes only.

7. We note that Figures SF2 and SF3 include images of participants in the study. 

Reviewers' comments:

Reviewer's Responses to Questions

**Comments to the Author**

1. Is the manuscript technically sound, and do the data support the conclusions?

Reviewer #1: Partly

Reviewer #2: Partly

Reviewer #3: No

2. Has the statistical analysis been performed appropriately and rigorously? 

Reviewer #1: Yes

Reviewer #2: I Don't Know

Reviewer #3: I Don't Know

3. Have the authors made all data underlying the findings in their manuscript fully available?

Reviewer #1: No

Reviewer #2: No

Reviewer #3: Yes

4. Is the manuscript presented in an intelligible fashion and written in standard English?

Reviewer #1: No

Reviewer #2: Yes

Reviewer #3: No

5. Review Comments to the Author

Reviewer #1: The research question is definitely of interest to readers in India and other countries in the tropics where snakebite is well and truly a occupational hazard. To the best of my knowledge a study along similar lines has not been carried out in India and the results of the study would help policy makers in bringing in changes to plug the gaps and deficiencies identified.

The research question interesting and the methodology fairly well charted.

The deficiency really is in the presentation of the findings and results.

- The abstract and introduction need to be written better and be representative of the actual research question. The introduction seems scattered and not to the point, going from numbers of snakebite to chronic sequelae which also includes death, venom constituents and effect to ASV. Restructuring the introduction and giving it a form should help make it read better. Would suggest adding some details on the tribal community and geography as well. The research question here is perceptions and awareness on snakebite envenomation, of the 202 respondents in the community it has not been explicitly stated whether all were tribals. The researchers could also specify whether the different tribes had different occupations vis a vis hunter, gatherer, farmer etc and their educational status.

62 higher than what, as only 13 states were assessed as a part of the million death study

82 unidentified- unknown

Lines 84,85 - it would be interpreted that death is a long term complication

Lines 94,95- Relevance of skin hypersensitivity in the context of the article

103-108 against the flow

Line 125 drinking water (potable water), would also imply sleeping is an activity?

Lines 139-141 – 4.5 % CFR included non venomous too (76% venomous and 24% non venomous)

Line 165 qualitative study, qualitative cannot be considered a study design, the study is cross sectional

Line 185 convenience sampling ? how representative would it be of the population studied

Line 207 it could be that they are unaware of their age

Line 345/443 peripheral health care workers, would be best sticking to frontline health workers

Line 395 how were the 18 out of 65 SC decided upon, what criteria was used to pick the same

Line 457 / 58 not clear what he intends to say is it that non venomous do not bite multiple times

The training for medical personnel seems not to have made much of a difference in improving knowledge of treating doctors

Was there an incentive given to the community responders, was it a convenience sampling and if so would it bias the results and could the result be considered as representative of the study population

The most important part of the result would be the perception and understanding of the subject of venomous snakebite among the community members presumably tribals, this for me, the qualitative study results are not clear. Themes and sub themes have been mentioned but only a few have been represented. If a table giving percentages of themes and subthemes were displayed the qualitative aspect would be much clearer. The qualitative analysis needs better representation both by way of themes and subthemes and its relevance to the community studied.

I take this opportunity to wish the researchers the best and hope they continue their research in this much neglected field.

Reviewer #2: Abstract

• Line 35 – In-depth-interviews are to be conducted using interview guide, not using pre-tested questionnaire.

Methodology & Results

• As the study is of qualitative nature, authors need to refer to COREQ checklist (http://cdn.elsevier.com/promis_misc/ISSM_COREQ_Checklist.pdf) and report the data analysis and study results.

Reviewer #3: 1. The area of this research is important for India where more than 50% of global snakebite deaths are happening there. A recent study published in E-life by Million Death Study collaborates finds 58k deaths and 1.1 to 1.7 million bites annually in India and one million deaths were in last 20 years. I learnt from a snakebite advocacy group meeting at ICMR, a substantial amount of expired unused anti venom vials throwing into trash without use in every year while many thousands die without treatments. This is a problem of either people not aware of (or trust) anti venom treatment in hospitals or physicians were not trained/confident enough in clinical management of snakebite cases or both.

2. According to the authors, the objective of this study is to test the awareness about prevention strategies, knowledge of first aids and treatment options among community; Snakebite patient management among snake handlers, healers, healthcare workers and medical personnel. Another set objective is to train the healthcare workers, a significant part of the manuscript allocated for that purpose. I am not sure the latter is a research?

3. I have some doubt of technical aspect of the methods. As describe in the manuscript this study is a qualitative research based on focus group discussions. At the same time, they use pretested questionnaires for data collection. To my knowledge, data were collected using recorded narratives in qualitative research instead use questionnaires. Information of these data are analyzed using standard software like Nvivo. It is difficult for me to comment anything about the method because this method is a mix of quantitative survey methods and qualitative research, which I am not familiar.

4. Commenting of statistical analysis and presentation, tables and in figures are very poorly presented. In particularly figures are powerful tools to exhibit study outcomes. It is difficult to grasp the messages going to reflect from figures and they should need significant improvements to bring into the journal standard.

5. Full of many unwanted stories, inclusion of unrelated stuffs, repeating the same information in many places, confusing non-standard English wordings and confusing sentences etc. are in everywhere of this manuscript and very hard to follow for me to read and understand about 60 pages of the manuscript.

6. Research reporting also seen some professional bias. Researchers as the allopathic medical personnel ask the traditional healers to disclose their treatments at focus group meetings in front of others. Researchers complaining they rejected to explain their treatment methods to them. I think it is not relevant for this study and these researchers don’t have a common sense about other professions.

7. I am sorry all my comments are negative here. It does not mean this manuscript does not have anything good. I didn’t spend my time to go too much deep commenting for improving the manuscript because so many technical flaws are already there.

6. PLOS authors have the option to publish the peer review history of their article (what does this mean?). If published, this will include your full peer review and any attached files.

Reviewer #1: No

Reviewer #2: No

Reviewer #3: No

---

## [Author Response · Author response to Decision Letter 0]

27 Feb 2021

Dear PLOS ONE Editorial Staff, 

Subject: Point by point response to reviewers and editors comments for the manuscript “Perceptions, awareness on snakebite envenomation among the tribal community and health care providers of Dahanu block, Palghar district in Maharashtra, India” 

Please find below our point-by-point response to comments of the academic editor and reviewers. Many thanks for your efforts on this manuscript.

Editors Comments 

Resposne: Thank you. We have tried our level best to format our manuscript according to the PLOS ONE’s style requirements. However, if there are any omissions, please point them out. We will continue to make corrections.

2. Please include your tables as part of your main manuscript and remove the individual files. Please note that supplementary tables should remain uploaded as separate "supporting information" files.

Resposne: Tables are included as part of main manuscript and supplementary tables are submitted as supporting information. 

Resposne: The manuscript has been edited for language usage, spelling, and grammar as per the recommendations. 

4. In the Methods, please discuss whether and how the questionnaire was validated and/or pre-tested. If these did not occur, please provide the rationale for not doing so.

Resposne: Thank you for the suggestion. We have added additional information about the questionnaire including validation and pre-testing. This information has been added to the ‘Data collection Tools’ sub-section of the “methods” section (Page No. 08; Line no. 170).

5. Thank you for stating the following in the Funding Section of your manuscript:

"RG is an awardee of the DBT

 Wellcome India alliance clinical and public health intermediate fellowship (Grant no.

IA/CPHI/18/1/503933)."

"Initials of the authors who received Funding : RG, SM

Grant numbers awarded :

Tribal Health Research Forum, Indian Council of Medical Research and Department of Health Research, Ministry of Health and Family Welfare, Government of India for financial assistance (NIRRH/MS/RA/886/03/2020)

The full name of each funder

URL of each funder website

www.icmr.nic.in

Did the sponsors or funders play any role in the study design, data collection and analysis, decision to publish, or preparation of the manuscript?

NO"

Resposne: Thanks for pointing out the problem.

 We have deleted the statement RG is an awardee of the DBT

 Wellcome India alliance clinical and public health intermediate fellowship (Grant no.

IA/CPHI/18/1/503933) from the acknowledgement section. 

6. We note that Figure 1 in your submission contain map images which may be copyrighted. All PLOS content is published under the Creative Commons Attribution License (CC BY 4.0), which means that the manuscript, images, and Supporting Information files will be freely available online, and any third party is permitted to access, download, copy, distribute, and use these materials in any way, even commercially, with proper attribution. For these reasons, we cannot publish previously copyrighted maps or satellite images created using proprietary data, such as Google software (Google Maps, Street View, and Earth). For more information, see our copyright guidelines: http://journals.plos.org/plosone/s/licenses-and-copyright.

6.1. You may seek permission from the original copyright holder of Figure 1 to publish the content specifically under the CC BY 4.0 license. 

6.2. If you are unable to obtain permission from the original copyright holder to publish these figures under the CC BY 4.0 license or if the copyright holder’s requirements are incompatible with the CC BY 4.0 license, please either i) remove the figure or ii) supply a replacement figure that complies with the CC BY 4.0 license. Please check copyright information on all replacement figures and update the figure caption with source information. If applicable, please specify in the figure caption text when a figure is similar but not identical to the original image and is therefore for illustrative purposes only.

Response: Thanks for your detailed guidance. We requested the copyright agency for the same, and we received “Email consent approval” to use the map (copy attached as Other File). 

7. We note that Figures SF2 and SF3 include images of participants in the study. 

Response: Thank you for guidance. We have removed the SF2 and SF3. 

Resposne to Reviewers comments 

Reviewer #1: 

The research question is definitely of interest to readers in India and other countries in the tropics where snakebite is well and truly a occupational hazard. To the best of my knowledge a study along similar lines has not been carried out in India and the results of the study would help policy makers in bringing in changes to plug the gaps and deficiencies identified. The research question interesting and the methodology fairly well charted.

The deficiency really is in the presentation of the findings and results. 

The abstract and introduction need to be written better and be representative of the actual research question. The introduction seems scattered and not to the point, going from numbers of snakebite to chronic sequelae which also includes death, venom constituents and effect to ASV. Restructuring the introduction and giving it a form should help make it read better. Would suggest adding some details on the tribal community and geography as well. The research question here is perceptions and awareness on snakebite envenomation, of the 202 respondents in the community it has not been explicitly stated whether all were tribals. The researchers could also specify whether the different tribes had different occupations vis a vis hunter, gatherer, farmer etc and their educational status.

Response: We thank the reviewer for suggestions to improve the strength of our manuscript. As per the reviewer’s recommendation, we have revised the abstract, introduction and also added additional information on details on the tribal community and geography. Additionally, the information different tribes and their occupations in study area is also added in the introduction section (Introduction - Page no. 06, line no. 124 -146). 

We greatly appreciate the reviewer’s critical comments and recommendations.

Line 62; higher than what, as only 13 states were assessed as a part of the million death study.

Response: The sentence is revised as follows: 

An earlier study conducted in India demonstrated the highest snakebite mortality rates per 100,000 with an average prevalence of 4.5 and varied between states from 3.0 (Maharashtra) to 6.2 (Andhra Pradesh), compared to (1.8) the rest of country”. (Manuscript page no. 03, line no 63-66). 

Line 82 unidentified- unknown

Response: 

As suggested, we have replaced the word unidentified with Unknown. (Manuscript page no. 04, line no 83). 

Lines 84,85 - it would be interpreted that death is a long term complication

Response:

We agree with the reviewer's suggestion. We have removed the word “Death”. (Manuscript page no. 04, line no 87)

Lines 94,95- Relevance of skin hypersensitivity in the context of the article

Response: We agree with the reviewer comment. The text on skin hypersensitivity is deleted. 

103-108 against the flow

Response: Thank you. The text (103-108) is deleted. 

Line 125 drinking water (potable water), would also imply sleeping is an activity?

Response: We agree with the reviewer's suggestion, changed the word drinking water with “Potable water. Removed the sleeping from activity. (Manuscript page no. 05, line no 103 -104) 

Lines 139-141 – 4.5 % CFR included non venomous too (76% venomous and 24% non venomous)

Response: The annual incidence of snakebite was 36 per 100,000 populations in Dahanu block. The total venomous bites were 110 of which five have died; So CFR is 4.5%. This does not include non-venomous bites. (Manuscript page no. 06, line no 118). 

Line 165 qualitative study, qualitative cannot be considered a study design, the study is cross sectional

Response: We thank the reviewer for suggestion on study design. As recommended, the study design is changed to cross sectional study”. (Manuscript page no.07, line no 149). 

Line 185 convenience sampling ? how representative would it be of the population studied

Response: The recruitment criteria was primarily focused on ensuring the representation of the diversity of the population in the study area. We conducted separate FGDs for men and women. We attempted to include older and younger participants in the community, community leaders and members, educated and uneducated participants. Majority of the FGD participants were representative of the tribal and non-tribal population in study area. This has been explained under methods section. 

Line 207 it could be that they are unaware of their age

Response: Yes we agree. It may be possible that the participants were unaware of their age. We have included this sentence in the revised manuscript (Page no. 10 , line no. 208-209).

Line 345/443 peripheral health care workers, would be best sticking to frontline health workers

Response: We thank for the suggestion. Peripheral health care workers is replaced by frontline health workers. Recommended changes are done throughout the Manuscript. 

Line 395 how were the 18 out of 65 SC decided upon, what criteria was used to pick the same

Response: The 18 out of 65 sub centers were selected by random selection method ( Page no.11, Line no. 239). 

Line 457 / 58 not clear what he intends to say is it that non venomous do not bite multiple times

Response: We have deleted the sentence line 457/58. 

The training for medical personnel seems not to have made much of a difference in improving knowledge of treating doctors

Response: We agree that the training of medical officers did not made much of a difference in improving of treating doctors. The possible reasons could be the educational background of the MOs who attended the training. Seventy two percent (18 out of 25) of the MOs were from Ayurveda discipline (non MBBS doctors) and only seven MOs were having qualifications of MBBS. Ayurveda is one of the most ancient systems of medicine serving people in the Indian subcontinent. Bachelor of Ayurvedic Medicine and Surgery (B.A.M.S.) is a professional degree in Ayurveda. We therefore conducted repeat training programs in the study area to improve the knowledge of MOs. 

Was there an incentive given to the community responders, was it a convenience sampling and if so would it bias the results and could the result be considered as representative of the study population

Response: No incentive was given to the community responders and the participation in FGD was voluntary. Convenience sampling method was used for selection of participants for focus group discussions. We included participants of both genders reflecting a wide range of ages, community leaders and members, older and younger participants representing the tribal and non-tribal population to avoid any bias. 

The most important part of the result would be the perception and understanding of the subject of venomous snakebite among the community members presumably tribals, this for me, the qualitative study results are not clear. Themes and sub themes have been mentioned but only a few have been represented. If a table giving percentages of themes and subthemes were displayed the qualitative aspect would be much clearer. The qualitative analysis needs better representation both by way of themes and subthemes and its relevance to the community studied.

I take this opportunity to wish the researchers the best and hope they continue their research in this much neglected field.

Response: We truly appreciate the reviewer’s in depth review of our manuscript and bringing out the shortcomings of our manuscript. This has helped us in improving the manuscript and presentation of the results of FGDs with more clarity. As per the recommendation, a table is prepared giving percentages of the subthemes. We take this opportunity to thank the reviewer for appreciation and motivating us to continue our research in this much neglected field. 

Major themes and Sub- themes of the study Frequency Percentage 

Major theme 1: Awareness and Knowledge about snakes

Names of venomous and non-venomous snakes 120 59.4%

Hiding places of snakes 106 52.5%

Identify venomous snakes 66 32.6 %

Season and timing of snakebites 32 15.8%

Major theme 2: Perception and health seeking behavior for snakebite

Beliefs related to snakebites 37 18.3%

Treatment at public health facility 67 33%

Treatment from traditional faith healers 24 11.8 %

Application of herbal products or home remedies 25 12.3%

Major theme 3: Knowledge on prevention, first aid and treatment for snakebite

Preventive measures awareness 35 17.3%

Use of tourniquet/ bandage 29 14.3%

Availability of snakebite treatment at public hospitals 17 8.4%

Transport to nearest public health facility 18 8.9%

Reviewer #2: Abstract

Line 35 – In-depth-interviews are to be conducted using interview guide, not using pre-tested questionnaire.

Response: We thank the reviewer for pointing out the mistake. We apologize for the inconvenience. We have made necessary changes in the manuscript.

Methodology & Results

As the study is of qualitative nature, authors need to refer to COREQ checklist (http://cdn.elsevier.com/promis_misc/ISSM_COREQ_Checklist.pdf) and report the data analysis and study results.

Response: Thank you for the recommendation of COREQ checklist. As per the recommendation, COREQ check list is included in the manuscript under method section (S1 text). (Manuscript Page no 07. line no. 158)

Reviewer #3: 

1. The area of this research is important for India where more than 50% of global snakebite deaths are happening there. A recent study published in E-life by Million Death Study collaborates finds 58k deaths and 1.1 to 1.7 million bites annually in India and one million deaths were in last 20 years. I learnt from a snakebite advocacy group meeting at ICMR, a substantial amount of expired unused anti venom vials throwing into trash without use in every year while many thousands die without treatments. This is a problem of either people not aware of (or trust) anti venom treatment in hospitals or physicians were not trained/confident enough in clinical management of snakebite cases or both.

Response: We thank the reviewer for acknowledging the importance of snakebite research for India. 

2. According to the authors, the objective of this study is to test the awareness about prevention strategies, knowledge of first aids and treatment options among community; Snakebite patient management among snake handlers, healers, healthcare workers and medical personnel. Another set objective is to train the healthcare workers, a significant part of the manuscript allocated for that purpose. I am not sure the latter is a research?

Response: As recommended, we have deleted the major text related to training of healthcare workers.

3. I have some doubt of technical aspect of the methods. As describe in the manuscript this study is a qualitative research based on focus group discussions. At the same time, they use pretested questionnaires for data collection. To my knowledge, data were collected using recorded narratives in qualitative research instead use questionnaires. Information of these data are analyzed using standard software like Nvivo. It is difficult for me to comment anything about the method because this method is a mix of quantitative survey methods and qualitative research, which I am not familiar.

Response: We apologize for not clearly presenting the qualitative results. As recommended by reviewer, the methodology and results are revised and reported as per COREQ checklist (Page no. 7, Line no 158). We did not use the software Nvivo and manual coding was done. 

4. Commenting of statistical analysis and presentation, tables and in figures are very poorly presented. In particularly figures are powerful tools to exhibit study outcomes. It is difficult to grasp the messages going to reflect from figures and they should need significant improvements to bring into the journal standard.

Response: As recommended we have substantially edited the manuscript, tables, and figures for improved presentation as per the journal standard.

5. Full of many unwanted stories, inclusion of unrelated stuffs, repeating the same information in many places, confusing non-standard English wordings and confusing sentences etc. are in everywhere of this manuscript and very hard to follow for me to read and understand about 60 pages of the manuscript.

Response: We apologize for the inconvenience caused to the reviewer. As recommended, we have revised the manuscript in light of the reviewer’s comments. 

6. Research reporting also seen some professional bias. Researchers as the allopathic medical personnel ask the traditional healers to disclose their treatments at focus group meetings in front of others. Researchers complaining they rejected to explain their treatment methods to them. I think it is not relevant for this study and these researchers don’t have a common sense about other professions.

Response: We did not conduct the FGD with traditional healers. We conducted interview of traditional healers individually and not in group. The reasons for not disclosing the treatment given by Traditional healers was explained in the manuscript. 

7. I am sorry all my comments are negative here. It does not mean this manuscript does not have anything good. I didn’t spend my time to go too much deep commenting for improving the manuscript because so many technical flaws are already there.

Response: We sincerely thank the reviewer for accepting the fact that our manuscript has merit. As per the recommendations, we have revised the manuscript substantially incorporating the suggestions of all the reviewers and editor.

---

## [Decision Letter · Decision Letter 1]

30 Mar 2021

PONE-D-20-33739R1

Perceptions, awareness on snakebite envenomation among the tribal community and health care providers of Dahanu block, Palghar district in Maharashtra, India

PLOS ONE

Dear Dr. Gajbhiye,

Thank you for submitting your manuscript to PLOS ONE. After careful consideration, we feel that it has merit but does not fully meet PLOS ONE’s publication criteria as it currently stands. Therefore, we invite you to submit a revised version of the manuscript that addresses the points raised during the review process.

We look forward to receiving your revised manuscript.

Kind regards,

Prof. Ritesh G. Menezes, M.B.B.S., M.D., Diplomate N.B.

Academic Editor

PLOS ONE

Journal Requirements:

Additional Academic Editor Comments:

- Few errors of English grammar and use persist in the revised manuscript. Copy-edit the manuscript before proceeding with the next submission.

- Avoid abuse of capital letters throughout the manuscript (examples: line 58 - replace ''Health care workers'' with ''health care workers''; line 50: replace ''Medical Officers'' with ''medical officers'')

- Abstract: ASHA? ANM? MPW?

- Line 113-Introduction: ASV - Provide the full form in the first place of use of the abbreviation in the main text.

-Line 146-Introduction: HCW - Provide the full form in the first place of use of the abbreviation in the main text. I am aware that the abbreviation/full-form is already being considered in the 'abstract' section.

- Line 152- Methods: ICU & OPD - Provide the full forms.

- Line 170 - Methods: PHC?

- Line 171 - Methods: MSW?

- Methods-Recruitment, Sampling & Data Collection: The use of the abbreviation DA in the text apparently seems to be confusing for a term rather than the initials of a co-author. Rewrite to avoid confusion as there are other abbreviations like FGD and HCW used in the same paragraph.

- Line 280 - Results: Would you prefer to substitute ''circulating snakes'' with a better phrase?

- Lines 306-312: The scientific name of Indian cobra is mentioned, but not that of the common krait, Russell's viper and saw scaled viper. All scientific names are mentioned along side the corresponding common names in Table 1. Therefore, for the sake of uniformity, avoid using the term Naja naja in this paragraph.

- Let the ''conclusions'' drawn be based on the data/observations/results of the present study. Provide separate paragraphs on ''recommendations/future directions'' and ''implications of the present study''.

- Address the minor revisions recommended by the reviewer(s).

Reviewers' comments:

Reviewer's Responses to Questions

**Comments to the Author**

1. If the authors have adequately addressed your comments raised in a previous round of review and you feel that this manuscript is now acceptable for publication, you may indicate that here to bypass the “Comments to the Author” section, enter your conflict of interest statement in the “Confidential to Editor” section, and submit your "Accept" recommendation.

Reviewer #1: All comments have been addressed

Reviewer #3: All comments have been addressed

2. Is the manuscript technically sound, and do the data support the conclusions?

Reviewer #1: Yes

Reviewer #3: Yes

3. Has the statistical analysis been performed appropriately and rigorously? 

Reviewer #1: Yes

Reviewer #3: I Don't Know

4. Have the authors made all data underlying the findings in their manuscript fully available?

Reviewer #1: Yes

Reviewer #3: Yes

5. Is the manuscript presented in an intelligible fashion and written in standard English?

Reviewer #1: Yes

Reviewer #3: No

6. Review Comments to the Author

Reviewer #1: The comments of the three reviewers have been taken into account by the authors and necessary revisions made. The manuscript makes for a much better read now and would be of interest to readers. The scientific / generic names of the snakes to be written correctly separating genus / species. Mention to be made of which echis species- carinatus or sochureki.

Wish the authors the best and I would recommend the article for publication, which I am sure would be of interest to a lot of readers.

Thanks for having given me the opportunity to review the manuscript.

regards

Jaideep

Dr Jaideep C Menon

Cardiologist, Amrita Institute of Medical Sciences

Kochi, Kerala

Reviewer #3: See attachment.

1. The area of this research is important for India where more than 50% of global snakebite

deaths are happening there. A recent study published in E-life by Million Death Study

collaborates finds 58k deaths and 1.1 to 1.7 million bites annually in India and one million deaths

were in last 20 years. I learnt from a snakebite advocacy group meeting at ICMR, a substantial

amount of expired unused anti venom vials throwing into trash without use in every year while

many thousands die without treatments. This is a problem of either people not aware of (or trust)

anti-venom treatment in hospitals or physicians were not trained/confident enough in clinical

management of snakebite cases or both.

Response: We thank the reviewer for acknowledging the importance of snakebite research for

India.

# Thanks.

2. According to the authors, the objective of this study is to test the awareness about prevention

strategies, knowledge of first aids and treatment options among community; Snakebite patient

management among snake handlers, healers, healthcare workers and medical personnel. Another

set objective is to train the healthcare workers, a significant part of the manuscript allocated for

that purpose. I am not sure the latter is a research?

Response: As recommended, we have deleted the major text related to training of healthcare

workers.

# Glad to see many improvements in the Introduction text. I like the new text from line 184 to 208 and is appropriate.

3. I have some doubt of technical aspect of the methods. As describe in the manuscript this study

is a qualitative research based on focus group discussions. At the same time, they use pretested

questionnaires for data collection. To my knowledge, data were collected using recorded

narratives in qualitative research instead use questionnaires. Information of these data are

analyzed using standard software like Nvivo. It is difficult for me to comment anything about the

method because this method is a mix of quantitative survey methods and qualitative research,

which I am not familiar.

Response: We apologize for not clearly presenting the qualitative results. As recommended by

reviewer, the methodology and results are revised and reported as per COREQ checklist (Page

no. 7, Line no 158). We did not use the software Nvivo and manual coding was done.

# Thanks for improvements in research method. It is now somewhat more comprehensible to me. Since my knowledge in qualitative research is limited, I leave it to other reviewers and the editor to take care.

4. Commenting of statistical analysis and presentation, tables and in figures are very poorly

presented. In particularly figures are powerful tools to exhibit study outcomes. It is difficult to

grasp the messages going to reflect from figures and they should need significant improvements

to bring into the journal standard.

Response: As recommended we have substantially edited the manuscript, tables, and figures for

improved presentation as per the journal standard.

# I am OK with the revision

5. Full of many unwanted stories, inclusion of unrelated stuffs, repeating the same information in

many places, confusing non-standard English wordings and confusing sentences etc. are in

everywhere of this manuscript and very hard to follow for me to read and understand about 60

pages of the manuscript.

Response: We apologize for the inconvenience caused to the reviewer. As recommended, we

have revised the manuscript in light of the reviewer’s comments.

# A good progress can be seen in revised manuscript. I think remaining repeating information and consistency of the text in the manuscript can take care by the editor with the help of the authors at the editorial process.

6. Research reporting also seen some professional bias. Researchers as the allopathic medical

personnel ask the traditional healers to disclose their treatments at focus group meetings in front

of others. Researchers complaining they rejected to explain their treatment methods to them. I

think it is not relevant for this study and these researchers don’t have a common sense about

other professions.

Response: We did not conduct the FGD with traditional healers. We conducted interview of

traditional healers individually and not in group. The reasons for not disclosing the treatment

given by Traditional healers was explained in the manuscript.

# I am OK with the revision of the text.

7. I am sorry all my comments are negative here. It does not mean this manuscript does not have

anything good. I didn’t spend my time to go too much deep commenting for improving the

manuscript because so many technical flaws are already there.

Response: We sincerely thank the reviewer for accepting the fact that our manuscript has merit.

As per the recommendations, we have revised the manuscript substantially incorporating the

suggestions of all the reviewers and editor.

# Thanks.

# Additional comments:

Throughout this manuscript uses the words ‘envenomation’ and ‘envenoming’ interchangeably. It looks authors have not given much attention to the difference between these 2 words. I think if the bite is an accident, then the best word would be the envenoming that WHO uses https://www.who.int/snakebites/disease/en/. Also I see another word “anti-snake venom”. The correct word should be the “anti-venom”. Refer same WHO and recent leading research articles on Snakebites.

7. PLOS authors have the option to publish the peer review history of their article (what does this mean?). If published, this will include your full peer review and any attached files.

Reviewer #1: **Yes: **Jaideep C Menon

Reviewer #3: No

---

## [Author Response · Author response to Decision Letter 1]

5 Apr 2021

RESPONSE TO REVIEWERS COMMENTS 

Dear PLOS ONE Editorial Staff, 

Many thanks for your efforts on this manuscript entitled “Perceptions, awareness on snakebite envenoming among the tribal community and health care providers of Dahanu block, Palghar district in Maharashtra, India”. 

Our point-by-point response to reviewer's and editor's comments are as follows. 

Journal Requirements:

Response: As suggested we have reviewed all the references quoted in the revised manuscript. We confirm that there are no retracted references cited in the manuscript. 

Academic editors comments 

1. -Few errors of English grammar and use persist in the revised manuscript. Copy-edit the manuscript before proceeding with the next submission. 

Response: Thanks for pointing out the errors of English grammer. We have edited the manuscript and tried our best to rectify the grammatical errors in the revised manuscript. 

2. - Avoid abuse of capital letters throughout the manuscript (examples: line 58 - replace ''Health care workers'' with ''health care workers''; line 50: replace ''Medical Officers'' with ''medical officers'')

Response: As suggested, we have done the changes throughout the revised manuscript. 

3. - Abstract: ASHA? ANM? MPW?

Response: As suggested we have included full forms of ASHA, ANM and MPW in the abstract section (line no: 50 & 51) and deleted in the main text (Line no: 232 & 233). 

4.- Line 113-Introduction: ASV - Provide the full form in the first place of use of the abbreviation in the main text.

Response: As recommended by reviewer 3, we have omitted the use of word anti-snake-venom (ASV) which is now replaced by “anti-venom” in the entire revised manuscript. 

5. -Line 146-Introduction: HCW - Provide the full form in the first place of use of the abbreviation in the main text. I am aware that the abbreviation/full-form is already being considered in the 'abstract' section.

Response: . As suggested, changes have been done in the revised manuscript. 

6. - Line 152- Methods: ICU & OPD - Provide the full forms.

Response: ICU & OPD full forms are added in the revised manuscript. Line No: 157,158 

7.- Line 170 - Methods: PHC? 

Response: PHC full form is added in the revised manuscript. Line No: 177 . 

8- Line 171 - Methods: MSW?

Response: MSW full form is added in the revised manuscript. Line No: 178. 

9.- Methods-Recruitment, Sampling & Data Collection: The use of the abbreviation DA in the text apparently seems to be confusing for a term rather than the initials of a co-author. Rewrite to avoid confusion as there are other abbreviations like FGD and HCW used in the same paragraph.

Response: Abbrivated authors names have been deleted in revised manuscript. Page No: 08 to10. 

10.- Line 280 - Results: Would you prefer to substitute ''circulating snakes'' with a better phrase?

Response: Accepted the suggestion, ''circulating snakes'' phrase is replaced with “commonly found snakes in the area”. Line No: 291. The changes are done throughout the manuscript. 

11. - Lines 306-312: The scientific name of Indian cobra is mentioned, but not that of the common krait, Russell's viper and saw scaled viper. All scientific names are mentioned along side the corresponding common names in Table 1. Therefore, for the sake of uniformity, avoid using the term Naja naja in this paragraph.

Response: Accepted the suggestion; the scientific name of Indian cobra is removed in the revised manuscript. Line no: 317.

12.- Let the ''conclusions'' drawn be based on the data/observations/results of the present study. Provide separate paragraphs on ''recommendations/future directions'' and ''implications of the present study''.

Response: As per the suggestions, we have revised the conclusion section. Also added separate paragraphs on recommendations and implications of the present study. Page no: 28, 29 line no: 593 & 607.

13. Address the minor revisions recommended by the reviewer(s).

Response : We have addressed the minor revisions recommended by the reviewers. 

Reviewer #1

The comments of the three reviewers have been taken into account by the authors and necessary revisions made. The manuscript makes for a much better read now and would be of interest to readers. The scientific / generic names of the snakes to be written correctly separating genus / species. Mention to be made of which echis species- carinatus or sochureki.

Wish the authors the best and I would recommend the article for publication, which I am sure would be of interest to a lot of readers.

Response: We sincerely thank the reviewer for critical comemnts and suggestions. Genus and species names of snakes have been corrected. The name of echis species i.e Echis carinatus is added in the revised manuscript. Table 1, page no: 15; Line no. 305. 

Reviewer # 3 

Glad to see many improvements in the Introduction text. I like the new text from line 184 to 208 and is appropriate. 

A good progress can be seen in revised manuscript. I think remaining repeating information and consistency of the text in the manuscript can take care by the editor with the help of the authors at the editorial process.

Response : Thanks for the feedback on the revised manuscript. 

# Additional comments

Throughout this manuscript uses the words ‘envenomation’ and ‘envenoming’ interchangeably. It looks authors have not given much attention to the difference between these 2 words. I think if the bite is an accident, then the best word would be the envenoming that WHO uses https://www.who.int/snakebites/disease/en/. Also I see another word “anti-snake venom”. The correct word should be the “anti-venom”. Refer same WHO and recent leading research articles on Snakebites.

Response: Thank you for the suggestion and sharing the link of WHO reference document for snakebite. In the revised manuscript, we have used the word envenoming and deleted the word envenomation. 

Similarly, the word “anti-snake venom” is replaced with “anti-venom” in the revised manuscript.

---

## [Decision Letter · Decision Letter 2]

6 May 2021

PONE-D-20-33739R2

Perceptions, awareness on snakebite envenoming among the tribal community and health care providers of Dahanu block, Palghar district in Maharashtra, India

PLOS ONE

Dear Dr. Gajbhiye,

Thank you for submitting your manuscript to PLOS ONE. After careful consideration, we feel that it has merit but does not fully meet PLOS ONE’s publication criteria as it currently stands. Therefore, we invite you to submit a revised version of the manuscript that addresses the points raised during the review process.

We look forward to receiving your revised manuscript.

Kind regards,

Prof. Ritesh G. Menezes, M.B.B.S., M.D., Diplomate N.B.

Academic Editor

PLOS ONE

Journal Requirements:

Reviewers' comments:

Reviewer's Responses to Questions

**Comments to the Author**

1. If the authors have adequately addressed your comments raised in a previous round of review and you feel that this manuscript is now acceptable for publication, you may indicate that here to bypass the “Comments to the Author” section, enter your conflict of interest statement in the “Confidential to Editor” section, and submit your "Accept" recommendation.

Reviewer #3: All comments have been addressed

2. Is the manuscript technically sound, and do the data support the conclusions?

Reviewer #3: Yes

3. Has the statistical analysis been performed appropriately and rigorously? 

Reviewer #3: Yes

4. Have the authors made all data underlying the findings in their manuscript fully available?

Reviewer #3: Yes

5. Is the manuscript presented in an intelligible fashion and written in standard English?

Reviewer #3: No

6. Review Comments to the Author

Reviewer #3: I am glad to see a reasonable progress in revised manuscript. Thanks, authors have well responded to my initial comments and no any technical issues from me.

Even tough I said no issues from me, manuscript needs a significant language improvement, condense the long descriptions squeeze into shorter, and eliminate all remaining repetitions and unrelated texts before publishing.

It is a serious challenge for editor and authors but would not be a big issue if you work seriously. Good luck

7. PLOS authors have the option to publish the peer review history of their article (what does this mean?). If published, this will include your full peer review and any attached files.

Reviewer #3: **Yes: **Wilson Suraweera

---

## [Author Response · Author response to Decision Letter 2]

21 May 2021

Response to Reviewer’s comments

Journal Requirements:

Response: 

We reviewed the references and confirm that the reference list is complete and correct. During the revision of the manuscript, the following references were deleted 

Reference no. 1,2,7,8, 33,38 

In the revised manuscript, the following references are added 

7. Sharma SK, Chappuis F, Jha N, Bovier PA, Loutan L, Koirala S. Impact of snake bites and determinants of fatal outcomes in southeastern Nepal. Am J Trop Med Hyg. 2004;71: 234–238.

31. Dey A, De JN. Traditional use of plants against snakebite in Indian subcontinent: a review of the recent literature. Afr J Tradit Complement Altern Med. 2011;9: 153–174. doi:10.4314/ajtcam.v9i1.20

Reviewer #3: I am glad to see a reasonable progress in revised manuscript. Thanks, authors have well responded to my initial comments and no any technical issues from me.

Even though I said no issues from me, manuscript needs a significant language improvement, condense the long descriptions squeeze into shorter, and eliminate all remaining repetitions and unrelated texts before publishing.

It is a serious challenge for editor and authors but would not be a big issue if you work seriously. Good luck

Response: As per the comments, we have edited the manuscript for language improvement. Also condensed the long sentences into shorter ones and removed the repetitions and unrelated text in the revised manuscript. We hope that the revised version of our manuscript is acceptable for publication.

---

## [Editor Report · Decision Letter 3]

22 Jul 2021

Perceptions, awareness on snakebite envenoming among the tribal community and health care providers of Dahanu block, Palghar district in Maharashtra, India

PONE-D-20-33739R3

Dear Dr. Gajbhiye,

We’re pleased to inform you that your manuscript has been judged scientifically suitable for publication and will be formally accepted for publication once it meets all outstanding technical requirements.

Kind regards,

Prof. Ritesh G. Menezes, M.B.B.S., M.D., Diplomate N.B.

Academic Editor

PLOS ONE

---

## [Editor Report · Acceptance letter]

26 Jul 2021

PONE-D-20-33739R3 

Perceptions, Awareness on Snakebite Envenoming  among the Tribal Community and Health Care providers of Dahanu Block, Palghar District in Maharashtra, India 

Dear Dr. Gajbhiye:

I'm pleased to inform you that your manuscript has been deemed suitable for publication in PLOS ONE. Congratulations! Your manuscript is now with our production department. 

Kind regards, 

on behalf of

Prof. Dr. Ritesh G. Menezes 

Academic Editor

PLOS ONE